# Why Did Urban Exodus Occur during the COVID-19 Pandemic from the Perspective of Residential Preference of Each Type of Household? Case of Japanese Metropolitan Areas

**Miyu Komaki, Haruka Kato *** and **Daisuke Matsushita**

Department of Housing and Environmental Design, Graduate School of Human Life and Ecology, Osaka Metropolitan University, Osaka 5588585, Japan
* Correspondence: haruka-kato@omu.ac.jp; Tel.: +81-6-6605-2823

**Abstract:** The background of this study is the urban exodus that occurred in Japanese metropolitan areas. The research question of this study is about the reasons why the urban exodus occurred in Japanese metropolitan areas. For the analysis, the objective of this study is to clarify the residential preferences of each household type in relation to the urban exodus during the COVID-19 pandemic in Japanese metropolitan areas. The method of this study is a web questionnaire survey. The sample comprised 593 respondents who migrated from ordinance-designed cities to other municipalities in metropolitan areas between April 2020 and March 2022. In conclusion, this study elucidates that migrant household type as urban exodus is households whose eldest child had enrolled in elementary school or above. Regarding residential preferences, the household type changes the importance of community and environment, rather than the working arrangement. This result is novel and essential because it clarifies that the household type tends to place more importance on the quality of childcare environment, ties to communities, the presence of a large garden/balcony, and utilizing opportunities to experience the community, such as via trial migration support programs.

**Keywords:** urban exodus; COVID-19 pandemic; residential preference; household type; Japanese metropolitan areas; childcare environment; tie to communities; access to workplaces

## 1. Introduction

### 1.1. Background

The COVID-19 pandemic made urban centers unlivable due to lockdowns and the collapse of medical systems. In addition, the pandemic changed lifestyles and workstyles, such as introducing working and learning from home. Due to those social changes, a rapid increase in migration from urban centers was reported [1].

This urban phenomenon is known as the urban exodus. In urban planning, an urban exodus is defined as a large-scale population migration from urban centers [2]. It gained attention in metropolitan areas worldwide in the wake of the COVID-19 pandemic. For example, in China, large numbers of migrants were reported by 24 January 2020, during the first lockdown [3]. In the New York metropolitan area, it was reported that, in April 2020, about 20% of the population moved from urban centers to suburban areas [4]. In Italy, it was also reported that people living in urban centers moved to suburban areas rather than rural areas [5]. Additionally, in Spain, migration from urban centers to suburban areas in the second half of 2020 was reported [6]. In Japan, Kato et al. [7] also found that an urban exodus occurred in the Osaka metropolitan area during the summer of 2020.

The study's research question is about the reasons why the urban exodus occurred in Japanese metropolitan areas. For the analysis, it is necessary to clarify the residential preference by household of those who migrated during the COVID-19 pandemic. Regarding the reasons for the urban exodus, Fukuda [8] elucidated the relationship between population change, fear of infection, and workstyle by industries in the Tokyo metropolitan area.

Furthermore, Nathan et al. [9] commented on the possibility that the COVID-19 pandemic weakened the importance of access to workplaces in relation to residential preferences. On the other hand, Nathan et al. [9] commented that the pandemic strengthened the importance of "urban attractiveness" over residential preference. However, the specifics of "urban attractiveness" that migrants seek in suburban areas have not been clarified. For example, Kato et al. [7] found that, even in suburban areas, municipalities were divided into those chosen as migration destinations and those not selected. Therefore, it is necessary to clarify the specifics of "urban attractiveness" according to the household types, as Nathan et al. [9] pointed out. Because the COVID-19 pandemic has changed residents' lifestyles and work-styles, it is significant to elucidate the changes in residential preference in relation to the urban exodus. The results might contribute to developing effective migration policies for urban planners and administrators in the post-pandemic era. Furthermore, the results could contribute to forecasting whether the urban exodus will be a temporary or long-term urban phenomenon in the post-pandemic era.

*1.2. Purpose*

This study aims to clarify the residential preferences of each household type in relation to the urban exodus during the COVID-19 pandemic in Japanese metropolitan areas. This study analyzes households who migrated from urban centers to suburban areas from April 2020 to March 2022. During this period, initial behavior was excessive restraint of their home range in Japan [10,11]. Concerning the suppression, during the summer of 2020, it was found that a large population migrated from urban centers to suburban areas in the Osaka metropolitan area [7]. In regard to migration, some newspapers reported that young people who could work from home were interested in migrating out of the urban centers [12]. In addition, it was reported that university students lived in their hometowns rather than the metropolitan area because many classes were mainly conducted remotely [13]. However, there has not been an exhaustive survey targeting the social background of the actual migrants, such as their economic level and social belonging. Therefore, analyzing the households could clarify the reasons why the urban exodus occurred. This study analyzes the Osaka and Tokyo metropolitan areas in Japan.

The method of this study is a web questionnaire survey. The sample comprised 593 respondents who migrated from ordinance-designed cities to other municipalities in metropolitan areas between April 2020 and March 2022. This study analyzes the Osaka and Tokyo metropolitan areas because they were reported to have encountered the urban exodus due to the pandemic [7,8]. Furthermore, the populations of the Osaka and Tokyo metropolitan areas began to decline due to advanced aging [14]. Therefore, population change due to urban exodus has been a unique urban phenomenon in these metropolitan areas.

In this study, urban centers refer to ordinance-designed cities in Japan, and suburban areas refer to municipalities other than ordinance-designed cities. Ordinance-designated cities are municipalities in which the central government designates to be cities with a population of more than 500,000. Ordinance-designed cities have authority structures similar to those in prefectures, in relation to policy making [15]. This study's ordinance-designed cities are: Kyoto City, Osaka City, Sakai City, and Kobe City in the Osaka Metropolitan area; the Tokyo 23 wards, Saitama City, Chiba City, Yokohama City, Kawasaki City, and Sagamihara City in the Tokyo Metropolitan area. In 2020, the population of the Osaka and Tokyo metropolitan areas was approximately 57 million, which comprised 45% of the Japanese population [16]. In the metropolitan areas, the population of the ordinance-designed cities was 24.6 million, which comprised 43% of the metropolitan population [16]. In the ordinance-designed cities, there are 8.6 million employees, which comprises about 80% of workers in the cities [17]. In addition, many households live in owned houses (6 million households) as well as rental houses (5.5 million households) in the cities [18].

The study period is from April 2020 to March 2022—Japan's pandemic. The first infected case in Japan was in January 2020 [19]. Subsequently, states of emergency were declared multiple times between April 2020 and March 2022 [20].

### 1.3. Literature Review

Urban exodus was reported in many countries, which is a unique urban phenomenon that occurred during the COVID-19 pandemic [2–7]. In regard to the urban exodus, Breuillé et al. [21] clarified that the desire to migrate increased as the pandemic and the restrictive measures continued in France. This migration affects many types of residents, such as homeless people [22] and immigrants [23]. For many residents, the COVID-19 pandemic created new needs and desires for migration [24]. However, there is a research gap in that we do not know the reason why urban exodus occurred, which has already been clarified as an urban phenomenon. Nathan et al. [9] commented that the pandemic strengthened the importance of "urban attractiveness". However, the specifics of "urban attractiveness" have not been clarified. It is essential to elucidate the reasons for the urban exodus along with the target household types, which could contribute to the development of effective urban policies.

Therefore, this study referred to research methods that were conducted before the pandemic. Residential preferences have been studied since before the COVID-19 pandemic. Gustavus et al. [25] stated that the relevant factors in residential preferences are housing, working, school, and medical care. In addition, Fredrickson et al. [26] elucidated the importance of ties to communities where migrants live. Sekiguchi et al. [27] also pointed out that the essential factors of residential preferences are not only ties to communities, but also the living environment. Therefore, this study analyzed residential preferences, including community, environment, working, and housing.

In previous studies, it was also found that these residential preferences are influenced by household type based on the individual life cycle [28]. Dökmeci et al. [29] elucidated that young households prioritized residing in the neighborhood of relatives. On the other hand, Shimizu et al. [30] clarified that in Japan young households prioritized access to workplaces. In addition, Ono et al. [31] elucidated that households with children who worked together prioritized access to the workplaces of wives. These results suggest that access to workplaces were generally crucial for all households before the COVID-19 pandemic. In addition, Kim et al. [32] pointed out that households with children chose their place of residence based on a trade-off between access to workplaces and the richness of the natural environment. They also pointed out that households without children made the choice based on a trade-off between access to workplace and living environments such as open spaces [32]. This result suggests that residential preferences are influenced by the presence of children. Therefore, this study analyzed residential preferences according to household types.

In contrast to these previous studies conducted before the COVID-19 pandemic, the academic contribution of this study regarding the urban exodus is that it elucidates the residential preferences of each household that migrated during the pandemic. The results could clarify the "urban attractiveness" that Nathan et al. [9] pointed out. "Urban attractiveness" might no longer include access to workplaces because of the spread of work from home (WFH). Instead, some household types might prefer their ties to local communities, as Fredrickson et al. [26] pointed out. Because of the pandemic, these household types may prefer the richness of the natural environment, access to medical facilities, and the information environment [33]. In addition, Kang et al. [34] clarified that those who were aged 40 or older and living in a townhouse or a single-detached house were more likely to consider moving to suburban areas in South Korea during the pandemic. However, it is very difficult to conduct a sampling of people who actually migrated during the pandemic. This is because no statistics accurately identify the number and demographics of people who have actually migrated as the urban exodus. Furthermore, it is difficult to survey migrants due to the protection of their personal information. This

means that this study needs to conduct unusual and extraordinary sampling. Therefore, this study referred to the web-based questionnaire survey that Tsuboi et al. [35] conducted for the analysis. It could elucidate the characteristics of households that migrated from urban centers to suburban areas in the urban exodus during the pandemic.

### 1.4. Article Structure

This manuscript consists of 5 chapters based on the IMRaD: materials and methods in Section 2; results in Section 3; discussion in Section 4; and conclusion in Section 5. In Section 3, the 7 subsections are analyzed sequentially.

## 2. Materials and Methods

### 2.1. Web Questionnaire Survey

This study conducted a web questionnaire survey, which is summarized in Table 1. The research protocol was approved by the Ethics Committee of the Graduate School of Human Life and Ecology at Osaka Metropolitan University (Approval No. 22-25). There were 593 respondents who migrated from ordinance-designed cities to other municipalities in the Osaka and Tokyo metropolitan areas between April 2020 and March 2022. The sampling method adopted the exhaustive investigation that targeted those registered to participate in the web-based questionnaire. The sampling method was not stratified extraction using age groups because there are no statistics criteria that accurately identify the demographics of people who have actually migrated during the pandemic. Therefore, the exhaustive investigation of those who registered to participate in the web-based questionnaire was the most appropriate method, although limited in number. The web questionnaire survey was conducted from 6 to 8 September 2022. The survey collected respondents by their zip codes to extract respondents who had moved from ordinance-designed cities to other municipalities in the Osaka and Tokyo metropolitan areas. This study directed the distribution and collection of the survey to Rakuten Insight, Inc. This is because Rakuten Insight, Inc. has the highest number of panels among all online research companies in Japan [36]. Additionally, it has the most reliable mechanism to eliminate fraudulent respondents [36]. Many academic articles have used data derived from cooperation with Rakuten Insight, Inc. (Tokyo, Japan) [37–39].

**Table 1.** Summary of web questionnaire survey.

| | |
|---|---|
| Method: | Web questionnaire survey |
| The number of Samples: | 593 |
| Screening of Samples: | Respondents who migrated from ordinance-designed cities to other municipalities in the Osaka and Tokyo metropolitan areas between April 2020 and March 2022. |
| Date: | 6–8 September 2022 |
| Questions: | (1). Migration: triggers (MA), residential preferences (MA), priorities of residential preferences (SA), and government support programs (MA). <br> (2). Place attachment: Williams' place attachment index (SA). <br> (3). Working: frequency of WFH (SA) and company support programs (MA). <br> (4). Housing: housing type (SA). <br> (5). Attributes: gender (SA), age (SA), household type (SA), and occupation (SA). |

Notes: SA are single-answer questions, MA are multiple-answer questions.

The survey consisted of five question topics: migration, place attachment, work, housing, and attributes. The questionnaire had two types of questions: single-answer questions (SA) and multiple-answer questions (MA). Specifically, the migration questions consisted of triggers (MA), residential preference (MA), priorities of residential preferences (SA), and government support programs (MA). The questions regarding place attachment relate to the place attachment index (SA), developed by Williams et al. [40]. This index has been widely used in many academic articles [41–44]. The questions regarding working relate to the frequency of WFH (SA) and company support programs (MA). The housing

questions related to housing type (SA). The questions regarding attributes focused on gender (SA), age (SA), household type (SA), and occupation (SA). The priority, housing type, and place attachment were compared before and during the pandemic. Before the pandemic refers to the previous residence prior to migration, and during the pandemic means the current residence since migration.

### 2.2. Statistical Analysis

This study conducted a cross-analysis of household types and responses to questions. Specifically, the questions concerned migration, place attachment, working, housing, and attributes, as shown in Table 1. Household types are classified into seven categories: single-person households (SH), married couples' households (MCH), households in which the eldest child has not yet enrolled in kindergarten (HcneK), households in which the eldest child has enrolled in kindergarten (HceK), households in which the eldest child has enrolled in elementary school or above (HceEa), households with children and grandparents (Hcg), and others. Based on previous studies [28–32], this study analyzes based on household types rather than age groups. Cross-tabulations indicated numbers and percentages vertically. Cross-tabulations were conducted using Fisher's exact establishment test (Monte Carlo Estimate) and residual analysis. The *p*-value criteria were set at 5% and 1%. The quartile was set at two criteria, where $\pm1.96$ indicates a significant difference at the 5% level of *p*-values, and $\pm2.56$ indicates significant differences at the 1% level of *p*-values. For the statistical analysis, this study used the IBM SPSS 29.0 software.

The place attachment is measured separately from place identity (PI) and place dependence (PD) [40]. Principal component analysis categorized the twelve scale items into PI and PD. The confidence analysis analyzed Cronbach's alpha. A paired-sample t-test was conducted for PI and PD before and during the pandemic.

## 3. Results

Section 3 analyzed the seven subsections according to the five question topics: migration, place attachment, work, housing, and attributes, which are listed in Table 1. First, Section 3.1 analyzed the respondents' attributes of each household type. Based on the attribute, migration was analyzed in Section 3.2 (triggers for migration) and Section 3.3 (residential preferences for migration). In regard to the residential preferences in migration, place attachment was analyzed in Section 3.4, housing was analyzed in Section 3.5, and working was analyzed in Section 3.6 sequentially. Finally, Section 3.7 analyzed the government support program.

In each table, colors indicate the significant differences by residual analysis: light red tabs indicate the quartile over 1.96; red tabs indicate the quartile over 2.56; light blue tabs indicate the quartile under −1.96; and blue tabs indicate the quartile under −2.56.

### 3.1. Respondent Attributes

Section 3.1 analyzed the respondents' attributes. Table 2 indicates the cross-tabulation of household types: gender/occupation/address. In Table 2, the gender ratio was nearly the same: male respondents comprised 53% (n = 315) and female respondents comprised 46% (n = 278). However, SHs yielded a significantly higher number of male respondents at the 1% level. In addition, HcneK gave a significantly higher number of female respondents at the 1% level.

In regard to occupation, company-employed (with second jobs) respondents significantly differed by household type. The differences in occupation might be related to the genders present in each household type. Specifically, HceEas had a significantly higher number of company employees (with second jobs) at the 5% level. In addition, HcneKs had a significantly higher number of housewives/househusbands at the 1% level.

Respondents' addresses were nearly the same; respondents in Osaka metropolitan areas comprised 38 % (n = 224) and respondents in Tokyo metropolitan areas comprised 62% (n = 369). The respondents' addresses were not significantly different from Osaka and

Tokyo metropolitan areas. In particular, households with children (HcneKs, HceKs, HceEas, and Hcgs) were not significantly different in the results of residual analysis.

**Table 2.** Respondent Attributes.

| | | | (Total) | SH | | MCH | | HcneK | | HceK | HceEa | | Hcg | Other | | *p* |
|---|---|---|---|---|---|---|---|---|---|---|---|---|---|---|---|---|
| Gender | Male | N | 315 | 95 | ++ | 79 | | 38 | – | 14 | 67 | | 13 | 9 | – | ** |
| | | (%) | (53) | (16) | | (13) | | (6) | | (2) | (11) | | (2) | (2) | | |
| | Female | N | 278 | 55 | – | 78 | | 55 | ++ | 16 | 45 | | 8 | 21 | ++ | |
| | | (%) | (47) | (9) | | (13) | | (9) | | (3) | (8) | | (1) | (4) | | |
| Occupation | Company employees (with second jobs) | N | 68 | 20 | | 16 | | 5 | - | 6 | 19 | + | 2 | 0 | - | ** |
| | | (%) | (11) | (3) | | (3) | | (1) | | (1) | (3) | | (0) | (0) | | |
| | Company employees (no second job) | N | 262 | 75 | | 63 | | 37 | | 15 | 43 | | 11 | 18 | | |
| | | (%) | (44) | (13) | | (11) | | (6) | | (3) | (7) | | (2) | (3) | | |
| | Manager, self-employed, freelance | N | 47 | 16 | | 16 | | 4 | | 0 | 8 | | 1 | 2 | | |
| | | (%) | (8) | (3) | | (3) | | (1) | | (0) | (1) | | (0) | (0) | | |
| | Contract employees/Temporary employees | N | 25 | 11 | + | 5 | | 1 | | 1 | 4 | | 2 | 1 | | |
| | | (%) | (4) | (2) | | (1) | | (0) | | (0) | (1) | | (0) | (0) | | |
| | Part-time job employees | N | 46 | 5 | - | 18 | + | 5 | | 1 | 13 | | 2 | 2 | | |
| | | (%) | (8) | (1) | | (3) | | (1) | | (0) | (2) | | (0) | (0) | | |
| | Public servants | N | 23 | 5 | | 7 | | 4 | | 0 | 6 | | 1 | 0 | | |
| | | (%) | (4) | (1) | | (1) | | (1) | | (0) | (1) | | (0) | (0) | | |
| | Medical professionals | N | 19 | 5 | | 3 | | 8 | ++ | 1 | 1 | | 0 | 1 | | |
| | | (%) | (3) | (1) | | (1) | | (1) | | (0) | (0) | | (0) | (0) | | |
| | Housewives/Househusbands | N | 61 | 2 | – | 15 | | 28 | ++ | 6 | 9 | | 0 | 1 | | |
| | | (%) | (10) | (0) | | (3) | | (5) | | (1) | (2) | | (0) | (0) | | |
| | Students | N | 8 | 2 | | 1 | | 0 | | 0 | 4 | + | 1 | 0 | | |
| | | (%) | (1) | (0) | | (0) | | (0) | | (0) | (1) | | (0) | (0) | | |
| | Retiree/early retiree | N | 22 | 4 | | 12 | | 0 | - | 0 | 4 | | 0 | 2 | | |
| | | (%) | (4) | (1) | | (2) | | (0) | | (0) | (1) | | (0) | (0) | | |
| | Others | N | 12 | 5 | | 1 | | 1 | | 0 | 1 | | 1 | 3 | ++ | |
| | | (%) | (2) | (1) | | (0) | | (0) | | (0) | (0) | | (0) | (1) | | |
| Address | Osaka Metropolitan Area | N | 224 | 45 | - | 69 | + | 38 | | 10 | 46 | | 4 | 12 | | |
| | | (%) | (38) | (8) | | (12) | | (6) | | (2) | (8) | | (1) | (2) | | |
| | Tokyo Metropolitan Area | N | 369 | 105 | + | 88 | - | 55 | | 20 | 66 | | 17 | 18 | | |
| | | (%) | (62) | (18) | | (15) | | (9) | | (3) | (11) | | (3) | (3) | | |

Note: +: quartile > 1.96 (light red tab); ++: quartile > 2.56 (red tab); -: quartile < −1.96 (light blue tab); –: quartile < −2.56 (blue tab); **: *p* < 0.01; SH are single-person households; MCH are married couples' households; HcneK are households in which the eldest child has not yet enrolled in kindergarten; HceK are households in which the eldest child has enrolled in kindergarten; HceEa are households in which the eldest child has enrolled in elementary school or above; Hcg are households with children and grandparents.

### 3.2. Triggers for Migration

The previous section clarified the gender, occupation, and address of each household type. According to the household types, Section 3.2 analyzed the triggers for migration. Table 3 shows the cross-tabulation of household types and the triggers for migration. As a result, we see that the triggers for migration during the pandemic were influenced mainly by the pandemic. Specifically, the top triggers were as follows: changes in lifestyle (N = 166, 28%), changes in workstyle (N = 129, 22%), and the spread of the COVID-19 infection (N = 121, 20%). HceEas were significantly triggered by the spread of COVID-19 infection at the 1% level. The HceEas were also significantly triggered by children's schooling and parental caregiving at the 1% level. These results suggest that the spread of the COVID-19 infection may have encouraged HceEas to migrate to suburban areas, because HceEas were more likely to have concerns about their children's schooling and their parental caregiving.

On the other hand, SHs were significantly triggered by the change in employment/ career at the 1% level. In addition, MCHs and HcneKs were triggered significantly by childbirth. These results remained unchanged from those before the pandemic. This means that the pandemic did not influence SHs, MCHs, or HcneKs.

**Table 3.** Triggers for migration.

| | | (Total) | SH | MCH | | HcneK | | HceK | | HceEa | | Hcg | | Other | | p |
|---|---|---|---|---|---|---|---|---|---|---|---|---|---|---|---|---|
| Spread of the COVID-19 infection | N | 121 | 23 | 32 | | 18 | | 7 | | 35 | ++ | 2 | | 4 | | |
| | (%) | (20) | (4) | (5) | | (3) | | (1) | | (6) | | (0) | | (1) | | |
| Changes in sense of values | N | 92 | 30 | 28 | - | 8 | | 2 | | 19 | | 1 | | 4 | | |
| | (%) | (16) | (5) | (5) | | (1) | | (0) | | (3) | | (0) | | (1) | | |
| Changes in workstyle | N | 129 | 41 | 28 | | 16 | | 5 | | 30 | | 5 | | 4 | | |
| | (%) | (22) | (7) | (5) | | (3) | | (1) | | (5) | | (1) | | (1) | | |
| Changes in lifestyle | N | 166 | 37 | 46 | | 23 | | 10 | | 36 | | 5 | | 9 | | |
| | (%) | (28) | (6) | (8) | | (4) | | (2) | | (6) | | (1) | | (2) | | |
| Referral from acquaintances or friends | N | 33 | 12 | 5 | | 4 | | 2 | | 9 | | 0 | | 1 | | |
| | (%) | (6) | (2) | (1) | | (1) | | (0) | | (2) | | (0) | | (0) | | |
| Tiredness of daily life | N | 48 | 16 | 13 | | 2 | - | 2 | | 12 | | 1 | | 2 | | |
| | (%) | (8) | (3) | (2) | | (0) | | (0) | | (2) | | (0) | | (0) | | |
| Change in employment/career | N | 104 | 45 | ++ | 18 | - | 7 | – | 1 | - | 21 | | 7 | | 5 | | ** |
| | (%) | (18) | (8) | (3) | | (1) | | (0) | | (4) | | (1) | | (1) | | |
| Working in agriculture | N | 12 | 4 | 1 | | 2 | | 0 | | 4 | | 1 | | 0 | | |
| | (%) | (2) | (1) | (0) | | (0) | | (0) | | (1) | | (0) | | (0) | | |
| Job relocation | N | 63 | 20 | 14 | | 10 | | 5 | | 11 | | 0 | | 3 | | |
| | (%) | (11) | (3) | (2) | | (2) | | (1) | | (2) | | (0) | | (1) | | |
| Marriage | N | 92 | 7 | 46 | ++ | 26 | ++ | 4 | | 4 | – | 3 | | 2 | | ** |
| | (%) | (16) | (1) | (8) | | (4) | | (1) | | (1) | | (1) | | (0) | | |
| Childbirth | N | 52 | 0 | – | 3 | – | 32 | ++ | 13 | ++ | 4 | - | 0 | | 0 | | ** |
| | (%) | (9) | (0) | (1) | | (5) | | (2) | | (1) | | (0) | | (0) | | |
| Children's schooling | N | 16 | 3 | 0 | - | 2 | | 1 | | 10 | ++ | 0 | | 0 | | ** |
| | (%) | (3) | (1) | (0) | | (0) | | (0) | | (2) | | (0) | | (0) | | |
| Divorce | N | 11 | 4 | 1 | | 2 | | 1 | | 3 | | 0 | | 0 | | |
| | (%) | (2) | (1) | (0) | | (0) | | (0) | | (1) | | (0) | | (0) | | |
| Retirement | N | 30 | 5 | 15 | ++ | 1 | | 0 | | 7 | | 0 | | 2 | | * |
| | (%) | (5) | (1) | (3) | | (0) | | (0) | | (1) | | (0) | | (0) | | |
| Parental caregiving | N | 19 | 4 | 3 | | 0 | | 0 | | 10 | ++ | 1 | | 1 | | * |
| | (%) | (3) | (1) | (1) | | (0) | | (0) | | (2) | | (0) | | (0) | | |
| Returning from abroad | N | 5 | 1 | 1 | | 0 | | 0 | | 1 | | 2 | ++ | 0 | | |
| | (%) | (1) | (0) | (0) | | (0) | | (0) | | (0) | | (0) | | (0) | | |
| Others | N | 52 | 12 | 13 | | 2 | - | 3 | | 10 | | 4 | | 8 | ++ | ** |
| | (%) | (9) | (2) | (2) | | (0) | | (1) | | (2) | | (1) | | (1) | | |

Note:; ++: quartile > 2.56 (red tab); -: quartile < −1.96 (light blue tab); –: quartile < −2.56 (blue tab); *: $p < 0.05$; **: $p < 0.01$; SH are single-person households; MCH are married couples' households; HcneK are households in which the eldest child has not yet enrolled in kindergarten; HceK are households in which the eldest child has enrolled in kindergarten; HceEa are households in which the eldest child has enrolled in elementary school or above; Hcg are households with children and grandparents.

### 3.3. Residential Preferences for Migration

The previous section clarified that the spread of COVID-19 infection significantly triggered HceEas. In regard to the HceEas, Section 3.3 analyzed the residential preferences for migration. Table 4 shows the cross-tabulation of household types and prioritization of residential preferences before and during the pandemic. As a result, we see that the residential priorities differ between before and during the pandemic. Before the pandemic, the order of priority was as follows: working (N = 203, 34%) > community (N = 141, 24%) > housing (N = 128, 22%) > environment (N = 121, 20%). It was found that the priorities were the same across different households because there were no significant differences between household types. In contrast, the order of priority changed during the pandemic as follows: housing (N = 200, 34%) > community (N = 152, 26%) > environment (N = 146, 25%) > working (N = 95, 16%). Furthermore, there were significant differences between household types during the pandemic. Specifically, HceEas significantly prioritized the community at the 1% level. On the other hand, SHs placed significant importance on working at the 1% level, which was the same as before the pandemic. These results suggest that HceEas changed their priorities in relation to the community for migration, although SHs still recognized the importance of working.

**Table 4.** Prioritization of residential preferences.

|  |  |  | (Total) | SH | MCH | HcneK | HceK | HceEa | Hcg | Other | p |
|---|---|---|---|---|---|---|---|---|---|---|---|
| Before the pandemic | Community | N | 141 | 40 | 40 | 18 | 5 | 28 | 7 | 3 | |
|  |  | (%) | (24) | (7) | (7) | (3) | (1) | (5) | (1) | (1) | |
|  | Environment | N | 121 | 28 | 25 | 21 | 5 | 26 | 6 | 10 | |
|  |  | (%) | (20) | (5) | (4) | (4) | (1) | (4) | (1) | (2) | |
|  | Working | N | 203 | 53 | 53 | 35 | 15 | 30 | 4 | 13 | |
|  |  | (%) | (34) | (9) | (9) | (6) | (3) | (5) | (1) | (2) | |
|  | Housing | N | 128 | 29 | 39 | 19 | 5 | 28 | 4 | 4 | |
|  |  | (%) | (22) | (5) | (7) | (3) | (1) | (5) | (1) | (1) | |
| During the pandemic | Community | N | 152 | 33 | 34 | 27 | 8 | 41  ++ | 4 | 5 | |
|  |  | (%) | (26) | (6) | (6) | (5) | (1) | (7) | (1) | (1) | |
|  | Environment | N | 146 | 31 | 47 | 28 | 7 | 24 | 3 | 6 | |
|  |  | (%) | (25) | (5) | (8) | (5) | (1) | (4) | (1) | (1) | |
|  | Working | N | 95 | 42  ++ | 22 | 5  – | 3 | 14 | 4 | 5 | ** |
|  |  | (%) | (16) | (7) | (4) | (1) | (1) | (2) | (1) | (1) | |
|  | Housing | N | 200 | 44 | 54 | 33 | 12 | 33 | 10 | 14 | |
|  |  | (%) | (34) | (7) | (9) | (6) | (2) | (6) | (2) | (2) | |

Note: ++: quartile > 2.56 (red tab); –: quartile < −2.56 (blue tab); **: $p < 0.01$; SH are single-person households; MCH are married couples' households; HcneK are households in which the eldest child has not yet enrolled in kindergarten; HceK are households in which the eldest child has enrolled in kindergarten; HceEa are households in which the eldest child has enrolled in elementary school or above; Hcg are households with children and grandparents.

Table 5 shows the cross-tabulation of household types and detailed residential preferences during the pandemic. The detailed residential preferences are categorized as community, environment, working, and housing. The top residential preferences are as follows: the richness of the natural environment (N = 216, 36%), favorability of the communities (N = 192, 32%), the access to transit infrastructure (N = 188, 32%), and many rooms (N = 186, 31%). These results indicate that more households prioritized the richness of the natural environment and the favorability of the communities during the pandemic.

We now focus on significant differences between household types. It was found that households with children place a strong emphasis on the quality of childcare environment (HceEas, HceKs, and HcneKs). When assessed by household type, we see that HceEas placed significant importance on the tie to communities and the presence of a large garden/balcony at the 1% level. Furthermore, HceEas also significantly emphasized a quality of childcare environment at the 1% level.

Furthermore, HcneKs placed significant importance on the locality of the neighborhoods of relatives and the access to parents' homes. The result suggests that for HcneKs people who help raise children are more important than the ties to communities. Thus, households with children commonly placed importance on the quality of childcare environment. However, HceEas placed importance on the tie to communities, although HcneKs placed significance on the neighborhood of relatives and the access to parents' homes. These results indicate that residential preferences changed according to household type and the age of the eldest child.

*3.4. Place Attachment for Migration*

The previous section clarified that HceEas significantly changed their priorities from work to the community, such as ties to communities. Therefore, concerning the community, Section 3.4 analyzed the place attachment in relation to migration. Table 6 indicates the average score for PI and PD of each household types before and during the pandemic. In addition, the Cronbach α score indicates the reliability of the results for PI and PD. Table 6 also shows the results of the paired-sample *t*-tests. PI refers to the value of a particular setting for emotional–symbolic reasons, and PD refers to the value based on functional

(activity-related) factors [45]. Moore et al. [45] elucidated that higher PD leads to repeated visits, which leads to higher PI.

**Table 5.** Detailed residential preferences.

| | | | (Total) | SH | MCH | HcneK | HceK | HceEa | Hcg | Other | p |
|---|---|---|---|---|---|---|---|---|---|---|---|
| Community: | Favorability of communities | N | 192 | 47 | 65 ++ | 22 - | 8 | 37 | 5 | 8 | |
| | | (%) | (32) | (8) | (11) | (4) | (1) | (6) | (1) | (1) | |
| | Tie to communities | N | 66 | 11 | 20 | 9 | 1 | 22 ++ | 3 | 0 + | ** |
| | | (%) | (11) | (2) | (3) | (2) | (0) | (4) | (1) | (0) | |
| | Neighborhood of acquaintances | N | 97 | 28 | 28 | 15 | 3 | 20 | 1 | 2 | |
| | | (%) | (16) | (5) | (5) | (3) | (1) | (3) | (0) | (0) | |
| | Neighborhood of relatives | N | 88 | 15 | 8 – | 25 ++ | 6 | 21 | 8 ++ | 5 | ** |
| | | (%) | (15) | (3) | (1) | (4) | (1) | (4) | (1) | (1) | |
| | Return to place occupied before | N | 107 | 22 | 22 | 18 | 6 | 24 | 5 | 10 + | |
| | | (%) | (18) | (4) | (4) | (3) | (1) | (4) | (1) | (2) | |
| | The balance between urban and rural | N | 151 | 41 | 48 | 25 | 4 | 24 | 2 | 7 | |
| | | (%) | (25) | (7) | (8) | (4) | (1) | (4) | (0) | (1) | |
| | Delicious agricultural foods | N | 33 | 9 | 9 | 3 | 2 | 8 | 1 | 1 | |
| | | (%) | (6) | (2) | (2) | (1) | (0) | (1) | (0) | (0) | |
| | Richness of natural environment | N | 216 | 54 | 64 | 29 | 9 | 39 | 8 | 13 | |
| | | (%) | (36) | (9) | (11) | (5) | (2) | (7) | (1) | (2) | |
| Environment: | Quality of childcare environment | N | 123 | 6 – | 24 - | 36 ++ | 17 ++ | 36 ++ | 2 | 2 - | ** |
| | | (%) | (21) | (1) | (4) | (6) | (3) | (6) | (0) | (0) | |
| | Access to places to enjoy holidays | N | 116 | 26 | 32 | 21 | 10 + | 20 | 4 | 3 | |
| | | (%) | (20) | (4) | (5) | (4) | (2) | (3) | (1) | (1) | |
| | Access to medical and welfare facilities | N | 53 | 9 | 14 | 11 | 4 | 13 | 0 | 2 | |
| | | (%) | (9) | (2) | (2) | (2) | (1) | (2) | (0) | (0) | |
| | Access to parents' homes | N | 123 | 25 | 21 – | 30 ++ | 8 | 26 | 8 + | 5 | ** |
| | | (%) | (21) | (4) | (4) | (5) | (1) | (4) | (1) | (1) | |
| | Access to transit infrastructure | N | 188 | 54 | 50 | 31 | 8 | 31 | 6 | 8 | |
| | | (%) | (32) | (9) | (8) | (5) | (1) | (5) | (1) | (1) | |
| | Access to commercial facilities | N | 41 | 12 | 16 | 3 | 0 | 8 | 1 | 1 | |
| | | (%) | (7) | (2) | (3) | (1) | (0) | (1) | (0) | (0) | |
| | Traditional and elegant landscapes | N | 77 | 15 | 19 | 19 + | 4 | 11 | 1 | 8 + | |
| | | (%) | (13) | (3) | (3) | (3) | (1) | (2) | (0) | (1) | |
| | Fewer disasters | N | 61 | 9 - | 19 | 11 | 3 | 16 | 2 | 1 | |
| | | (%) | (10) | (2) | (3) | (2) | (1) | (3) | (0) | (0) | |
| | Less noise | N | 94 | 19 | 28 | 12 | 3 | 22 | 3 | 7 | |
| | | (%) | (16) | (3) | (5) | (2) | (1) | (4) | (1) | (1) | |
| | Good public safety | N | 125 | 26 | 38 | 18 | 9 | 21 | 7 | 6 | |
| | | (%) | (21) | (4) | (6) | (3) | (2) | (4) | (1) | (1) | |
| Working: | Access to workplaces | N | 175 | 61 ++ | 44 | 22 | 4 - | 34 | 4 | 6 | * |
| | | (%) | (30) | (10) | (7) | (4) | (1) | (6) | (1) | (1) | |
| | Availability of WFH | N | 152 | 33 | 41 | 20 | 4 | 34 | 9 | 11 | |
| | | (%) | (26) | (6) | (7) | (3) | (1) | (6) | (2) | (2) | |
| | Business opportunities | N | 32 | 6 | 8 | 6 | 0 | 10 | 2 | 0 | |
| | | (%) | (5) | (1) | (1) | (1) | (0) | (2) | (0) | (0) | |
| | Certainty of potential customers | N | 13 | 2 | 4 | 2 | 1 | 2 | 2 + | 0 | |
| | | (%) | (2) | (0) | (1) | (0) | (0) | (0) | (0) | (0) | |
| Housing: | Many rooms | N | 186 | 29 – | 55 | 34 | 11 | 38 | 8 | 11 | * |
| | | (%) | (31) | (5) | (9) | (6) | (2) | (6) | (1) | (2) | |
| | Large garden/balcony | N | 115 | 15 – | 28 | 17 | 8 | 33 ++ | 5 | 9 | * |
| | | (%) | (19) | (3) | (5) | (3) | (1) | (6) | (1) | (2) | |
| | Renovation possible | N | 55 | 12 | 13 | 11 | 0 | 15 | 2 | 2 | |
| | | (%) | (9) | (2) | (2) | (2) | (0) | (3) | (0) | (0) | |
| | Resistance to earthquakes | N | 76 | 9 – | 19 | 17 | 4 | 20 | 3 | 4 | * |
| | | (%) | (13) | (2) | (3) | (3) | (1) | (3) | (1) | (1) | |
| | Resistance to fires | N | 43 | 6 | 16 | 11 | 2 | 5 | 2 | 1 | |
| | | (%) | (7) | (1) | (3) | (2) | (0) | (1) | (0) | (0) | |
| | Others | N | 47 | 12 | 10 | 6 | 1 | 10 | 2 | 6 + | |
| | | (%) | (8) | (2) | (2) | (1) | (0) | (2) | (0) | (1) | |

Note: +: quartile > 1.96 (light red tab); ++: quartile > 2.56 (red tab); -: quartile < −1.96 (light blue tab); –: quartile < −2.56 (blue tab); *: $p < 0.05$; **: $p < 0.01$; SH are single-person households; MCH are married couples' households; HcneK are households in which the eldest child has not yet enrolled in kindergarten; HceK are households in which the eldest child has enrolled in kindergarten; HceEa are households in which the eldest child has enrolled in elementary school or above; Hcg are households with children and grandparents.

As a result, Table 6 shows that HceEas significantly differed in PD at the 1% level. HceEas gave a higher PD in relation to the place to which they immigrated. The results of HceEas might be related to the residential preferences regarding children's schooling and parental caregiving. HceEas showed no significant differences in PI, which favors the emotional–symbolic feeling. Therefore, HceEa prioritized new communities over familiar ones.

**Table 6.** Place attachment for migration location.

| | | | SH | | MCH | HcneK | HceK | | HceEa | | Hcg | Other | Cronbach α |
|---|---|---|---|---|---|---|---|---|---|---|---|---|---|
| Place Identity | Before the pandemic | (Ave.) | 0.09 | * | 0.04 | −0.23 | −0.13 | ** | 0.02 | | −0.02 | 0.12 | 0.92 |
| | During the pandemic | (Ave.) | −0.14 | | −0.02 | −0.17 | 0.55 | | 0.2 | | 0.05 | 0 | 0.91 |
| Place Dependence | Before the pandemic | (Ave.) | 0.08 | ** | 0.08 | −0.17 | −0.07 | ** | −0.04 | ** | 0.04 | −0.13 | 0.83 |
| | During the pandemic | (Ave.) | −0.13 | | 0.02 | −0.15 | 0.53 | | 0.21 | | −0.05 | −0.23 | 0.85 |

Note: *: $p$-value < 0.05 (light yellow tab); **: $p$-value < 0.01 (yellow tab); SH are single-person households; MCH are married couples' households; HcneK are households in which the eldest child has not yet enrolled in kindergarten; HceK are households in which the eldest child has enrolled in kindergarten; HceEa are households in which the eldest child has enrolled in elementary school or above; Hcg are households with children and grandparents.

A significant difference was also found in PI and PD between before and during the pandemic among SHs and HceKs. SHs gave lower PI and PD scores to migration location during the pandemic. On the other hand, HceKs gave higher PI and PD scores to the place of migration during the pandemic. The result suggests that SHs did not feel attachments to the place to which they migrated, but HceKs did. HceKs were more likely to move to places with high PI from the beginning. Even before the pandemic, access to parents' homes were prioritized by some household types that sought help with raising their children. The residential preference did not change among HceKs from before to during the pandemic.

*3.5. Housing Types*

In Section 3.3, it was found that HceEas placed significant importance on the presence of a large garden/balcony. Therefore, the following Section 3.5 analyzed housing types about migration. Table 7 shows the cross-tabulation of household types and housing types before and during the pandemic. Table 7 shows that those living in new detached houses for sale gave significantly different from before to during the pandemic. During the pandemic, HceEas were significantly more likely to live in new detached houses for sale at the 1% level. HceKs were also significantly more likely to live in new detached houses for sale at the 1% level. In addition, HcneKs and Hcg were significantly more likely to live in new detached houses for sale at the 5% level. HceEas, HceKs, and HcneKs were not significantly more likely to live in new detached houses for sale before the pandemic. For example, before the pandemic, HceEas suggested a significantly higher preference, at the 1% level, to live in public housing. The results suggest that households with children were likelier to live in new detached houses for sale, because they preferred the quality of childcare environment and the presence of large gardens/balconies.

**Table 7.** Housing types.

| | | | (Total) | SH | | MCH | HcneK | | HceK | | HceEa | | Hcg | | Other | | p |
|---|---|---|---|---|---|---|---|---|---|---|---|---|---|---|---|---|---|
| Before the pandemic: | New detached house for sale | N | 92 | 18 | | 25 | 12 | | 3 | | 24 | | 7 | + | 3 | | * |
| | | (%) | (16) | (3) | | (4) | (2) | | (1) | | (4) | | (1) | | (1) | | |
| | Used detached house for sale | N | 20 | 8 | | 6 | 2 | | 2 | | 2 | | 0 | | 0 | | |
| | | (%) | (3) | (1) | | (1) | (0) | | (0) | | (0) | | (0) | | (0) | | |
| | Used detached house for rent | N | 20 | 3 | | 2 | 5 | | 3 | + | 5 | | 1 | | 1 | | |
| | | (%) | (3) | (1) | | (0) | (1) | | (1) | | (1) | | (0) | | (0) | | |
| | New apartment for sale | N | 54 | 10 | | 18 | 6 | | 3 | | 13 | | 2 | | 2 | | |
| | | (%) | (9) | (2) | | (3) | (1) | | (1) | | (2) | | (0) | | (0) | | |
| | Used apartment for sale | N | 21 | 6 | | 5 | 2 | | 1 | | 7 | | 0 | | 0 | | |
| | | (%) | (4) | (1) | | (1) | (0) | | (0) | | (1) | | (0) | | (0) | | |
| | Apartment for rent | N | 322 | 84 | | 80 | 59 | | 17 | | 48 | − | 10 | | 24 | ++ | |
| | | (%) | (54) | (14) | | (13) | (10) | | (3) | | (8) | | (2) | | (4) | | |
| | Public housing | N | 16 | 1 | | 6 | 0 | | 0 | | 8 | ++ | 1 | | 0 | | |
| | | (%) | (3) | (0) | | (1) | (0) | | (0) | | (1) | | (0) | | (0) | | |
| | Employee's house | N | 41 | 17 | + | 14 | 5 | | 1 | | 4 | | 0 | | 0 | | |
| | | (%) | (7) | (3) | | (2) | (1) | | (0) | | (1) | | (0) | | (0) | | |
| | Others | N | 7 | 3 | | 1 | 2 | | 0 | | 1 | | 0 | | 0 | | |
| | | (%) | (1) | (1) | | (0) | (0) | | (0) | | (0) | | (0) | | (0) | | |
| During the pandemic: | New detached house for sale | N | 153 | 11 | − | 35 | 33 | + | 14 | ++ | 40 | ++ | 10 | + | 10 | | ** |
| | | (%) | (26) | (2) | | (6) | (6) | | (2) | | (7) | | (2) | | (2) | | |
| | Used detached house for sale | N | 61 | 14 | | 22 | 3 | - | 1 | | 14 | | 4 | | 3 | | |
| | | (%) | (10) | (2) | | (4) | (1) | | (0) | | (2) | | (1) | | (1) | | |
| | Used detached house for rent | N | 39 | 8 | | 10 | 8 | | 4 | | 5 | | 1 | | 3 | | |

**Table 7.** *Cont.*

|  |  | (Total) | SH | | MCH | HcneK | | HceK | HceEa | | Hcg | | Other | | p |
|---|---|---|---|---|---|---|---|---|---|---|---|---|---|---|---|
| New apartment for sale | (%) | (7) | (1) | | (2) | (1) | | (1) | (1) | | (0) | | (1) | | |
| | N | 51 | 6 | - | 14 | 14 | + | 2 | 13 | | 1 | | 1 | | |
| Used apartment for sale | (%) | (9) | (1) | | (2) | (2) | | (0) | (2) | | (0) | | (0) | | |
| | N | 33 | 2 | – | 11 | 7 | | 0 | 11 | + | 1 | | 1 | | |
| Rent apartment | (%) | (6) | (0) | | (2) | (1) | | (0) | (2) | | (0) | | (0) | | |
| | N | 214 | 92 | ++ | 59 | 24 | - | 7 | 21 | – | 2 | – | 9 | | |
| Public housing | (%) | (36) | (16) | | (10) | (4) | | (1) | (4) | | (0) | | (2) | | |
| | N | 8 | 1 | | 1 | 0 | | 1 | 3 | | 1 | | 1 | | |
| Employee's house | (%) | (1) | (0) | | (0) | (0) | | (0) | (1) | | (0) | | (0) | | |
| | N | 27 | 15 | ++ | 4 | 4 | | 0 | 4 | | 0 | | 0 | | |
| Others | (%) | (5) | (3) | | (1) | (1) | | (0) | (1) | | (0) | | (0) | | |
| | N | 7 | 1 | | 1 | 0 | | 1 | 1 | | 1 | | 2 | ++ | |
| | (%) | (1) | (0) | | (0) | (0) | | (0) | (0) | | (0) | | (0) | | |

Note: +: quartile > 1.96 (light red tab); ++: quartile > 2.56 (red tab); -: quartile < −1.96 (light blue tab); –: quartile < −2.56 (blue tab); *: $p < 0.05$; **: $p < 0.01$; SH is single-person households; MCH is married couples' households; HcneK is households in which the eldest child has not yet enrolled in kindergarten; HceK is households in which the eldest child has enrolled in kindergarten; HceEa is households in which the eldest child has enrolled in elementary school or above; Hcg is households with children and grandparents.

### 3.6. Working Style

In Section 3.3, it was found that HceEas significantly changed their priorities from work, although all household types prioritized work before the pandemic. In regard to the work, Section 3.6 analyzed the working styles in relation to migration. Table 8 shows the cross-tabulation of household types and the ratio of office work (OW) to WFH between before and during the pandemic. As a result, we see no significant differences between households before the pandemic. Before the pandemic, a ratio of OW:WFH = 100:0% was the most common response for all generations (N = 388, 65%). This means most households worked from the office, with no WFH. However, there were significant differences by household type at the 1% level during the pandemic. We found that the response OW:WFH = 80:20% was significantly more prominent in HceEas at the 1% level. For HcneKs, OW:WFH = 0:100% was significantly more prominent at the 1% level. In addition, for SHs, OW:WFH = 50:50% was significantly more prominent at the 5% level. This means that workstyles have changed by household types in the wake of the pandemic.

**Table 8.** Ratio of office work to work from home.

|  |  |  | (Total) | SH | | MCH | HcneK | | HceK | HceEa | | Hcg | | Other | p |
|---|---|---|---|---|---|---|---|---|---|---|---|---|---|---|---|
| Before the pandemic: | OW:WFH = 0:100% | N | 65 | 22 | | 17 | 9 | | 3 | 9 | | 4 | | 1 | |
| | | (%) | (11) | (4) | | (3) | (2) | | (1) | (2) | | (1) | | (0) | |
| | OW:WFH = 20:80% | N | 20 | 6 | | 3 | 2 | | 1 | 6 | | 1 | | 1 | |
| | | (%) | (3) | (1) | | (1) | (0) | | (0) | (1) | | (0) | | (0) | |
| | OW:WFH = 50:50% | N | 29 | 6 | | 9 | 5 | | 1 | 7 | | 1 | | 0 | |
| | | (%) | (5) | (1) | | (2) | (1) | | (0) | (1) | | (0) | | (0) | |
| | OW:WFH = 80:20% | N | 27 | 10 | | 5 | 4 | | 3 | 4 | | 1 | | 0 | |
| | | (%) | (5) | (2) | | (1) | (1) | | (1) | (1) | | (0) | | (0) | |
| | OW:WFH = 100:0% | N | 388 | 96 | | 101 | 61 | | 19 | 74 | | 13 | | 24 | |
| | | (%) | (65) | (16) | | (17) | (10) | | (3) | (12) | | (2) | | (4) | |
| | Not working | N | 64 | 10 | | 22 | 12 | | 3 | 12 | | 1 | | 4 | |
| | | (%) | (11) | (2) | | (4) | (2) | | (1) | (2) | | (0) | | (1) | |
| During the pandemic: | OW:WFH = 0:100% | N | 110 | 16 | – | 30 | 34 | ++ | 6 | 15 | | 2 | | 7 | ** |
| | | (%) | (19) | (3) | | (5) | (6) | | (1) | (3) | | (0) | | (1) | |
| | OW:WFH = 20:80% | N | 99 | 29 | | 26 | 18 | | 6 | 11 | - | 3 | | 6 | |
| | | (%) | (17) | (5) | | (4) | (3) | | (1) | (2) | | (1) | | (1) | |
| | OW:WFH = 50:50% | N | 60 | 22 | + | 14 | 5 | | 2 | 13 | | 1 | | 3 | |
| | | (%) | (10) | (4) | | (2) | (1) | | (0) | (2) | | (0) | | (1) | |
| | OW:WFH = 80:20% | N | 69 | 16 | | 16 | 7 | | 2 | 23 | ++ | 2 | | 3 | |
| | | (%) | (12) | (3) | | (3) | (1) | | (0) | (4) | | (0) | | (1) | |
| | OW:WFH = 100:0% | N | 59 | 15 | | 11 | 10 | | 5 | 13 | | 1 | | 4 | |
| | | (%) | (10) | (3) | | (2) | (2) | | (1) | (2) | | (0) | | (1) | |
| | Not working | N | 196 | 52 | | 60 | 19 | – | 9 | 37 | | 12 | + | 7 | |
| | | (%) | (33) | (9) | | (10) | (3) | | (2) | (6) | | (2) | | (1) | |

Note: +: quartile > 1.96 (light red tab); ++: quartile > 2.56 (red tab); -: quartile < −1.96 (light blue tab); –: quartile < −2.56 (blue tab); **: $p < 0.01$; SH are single-person households; MCH are married couples' households; HcneK are households in which the eldest child has not yet enrolled in kindergarten; HceK are households in which the eldest child has enrolled in kindergarten; HceEa are households in which the eldest child has enrolled in elementary school or above; Hcg are households with children and grandparents.

This section assessed the households that used company support programs during the pandemic, as shown in Table 9. HceEas were more likely to use the flextime working program at the 1% level. HceEas were also significantly more likely to use job-type employment programs and second job support programs at the 5% level. The results suggest that HceEas were more likely than other households to implement WFH and use the company's programs more actively. Furthermore, HcneKs and HceKs were significantly more likely, at the 1% level, to use childcare leave support programs.

**Table 9.** Company support systems.

| | | (Total) | SH | | MCH | | HcneK | | HceK | | HceEa | | Hcg | Other | | p |
|---|---|---|---|---|---|---|---|---|---|---|---|---|---|---|---|---|
| WFH | N | 236 | 58 | | 50 | - | 40 | | 11 | | 48 | | 10 | 19 | ++ | * |
| | (%) | (40) | (10) | | (8) | | (7) | | (2) | | (8) | | (2) | (3) | | |
| Flextime working programs | N | 184 | 44 | | 39 | - | 31 | | 8 | | 47 | ++ | 6 | 9 | | |
| | (%) | (31) | (7) | | (7) | | (5) | | (1) | | (8) | | (1) | (2) | | |
| Job-type employment programs | N | 63 | 18 | | 10 | - | 12 | | 3 | | 19 | + | 0 | 1 | | * |
| | (%) | (11) | (3) | | (2) | | (2) | | (1) | | (3) | | (0) | (0) | | |
| Childcare leave support programs | N | 85 | 8 | − | 11 | − | 35 | ++ | 13 | ++ | 14 | | 1 | 3 | | ** |
| | (%) | (14) | (1) | | (2) | | (6) | | (2) | | (2) | | (0) | (1) | | |
| Second job support programs | N | 55 | 11 | | 16 | | 8 | | 1 | | 16 | + | 1 | 2 | | |
| | (%) | (9) | (2) | | (3) | | (1) | | (0) | | (3) | | (0) | (0) | | |
| Not applicable | N | 223 | 65 | | 75 | ++ | 24 | − | 9 | | 32 | − | 8 | 10 | | ** |
| | (%) | (38) | (11) | | (13) | | (2) | | (2) | | (5) | | (1) | (2) | | |

Note: +: quartile > 1.96 (light red tab); ++: quartile > 2.56 (red tab); -: quartile < −1.96 (light blue tab); –: quartile < −2.56 (blue tab); *: *p* < 0.05; **: *p* < 0.01; SH are single-person households; MCH are married couples' households; HcneK are households in which the eldest child has not yet enrolled in kindergarten; HceK are households in which the eldest child has enrolled in kindergarten; HceEa are households in which the eldest child has enrolled in elementary school or above; Hcg are households with children and grandparents.

### 3.7. Government Support Programs

The previous sections clarified the residential preferences of each household from many viewpoints. In regard to migration as the urban exodus, what policies would be effective in supporting this migration? In response to the question, Section 3.7 analyzed the use of government support programs in relation to migration. Table 10 shows that HcneKs, HceKs, and HceEas were significantly more likely to use childcare support programs at the 1% level. Among them, HceEas were significantly more likely to use the trial migration experience programs and the welfare support programs at the 1% level. The results suggest that HceEas identified communities into which they wanted to migrate using the trial migration experience program. Furthermore, this result indicates that HceEas were more likely to use welfare support programs because they were concerned about parental caregiving.

**Table 10.** Government support programs.

| | | (Total) | SH | | MCH | | HcneK | | HceK | | HceEa | | Hcg | Other | | p |
|---|---|---|---|---|---|---|---|---|---|---|---|---|---|---|---|---|
| Providing information on vacant houses | N | 152 | 49 | + | 40 | | 18 | | 6 | | 29 | | 4 | 6 | | |
| | (%) | (26) | (8) | | (7) | | (3) | | (1) | | (5) | | (1) | (1) | | |
| Trial migration experience programs | N | 70 | 15 | | 15 | | 11 | | 3 | | 22 | ++ | 2 | 2 | | |
| | (%) | (12) | (3) | | (3) | | (2) | | (1) | | (4) | | (0) | (0) | | |
| Migration support subsidy programs | N | 115 | 29 | | 33 | | 15 | | 6 | | 25 | | 3 | 4 | | |
| | (%) | (19) | (5) | | (6) | | (3) | | (1) | | (4) | | (1) | (1) | | |
| Employment support programs | N | 57 | 22 | + | 13 | | 5 | | 3 | | 12 | | 2 | 0 | | |
| | (%) | (10) | (4) | | (2) | | (1) | | (1) | | (2) | | (0) | (0) | | |
| Childcare support programs | N | 156 | 6 | − | 21 | − | 62 | ++ | 20 | ++ | 41 | ++ | 5 | 1 | − | ** |
| | (%) | (26) | (1) | | (4) | | (10) | | (3) | | (7) | | (1) | (0) | | |
| Welfare support programs | N | 71 | 14 | | 19 | | 5 | - | 2 | | 22 | ++ | 4 | 5 | | * |
| | (%) | (12) | (2) | | (3) | | (1) | | (0) | | (4) | | (1) | (1) | | |
| Others | N | 156 | 55 | ++ | 57 | ++ | 5 | − | 2 | - | 15 | − | 7 | 15 | ++ | ** |
| | (%) | (26) | (9) | | (10) | | (1) | | (0) | | (3) | | (1) | (3) | | |

Note: +: quartile > 1.96 (light red tab); ++: quartile > 2.56 (red tab); -: quartile < −1.96 (light blue tab); –: quartile < −2.56 (blue tab); *: *p* < 0.05; **: *p* < 0.01; SH are single-person households; MCH are married couples' households; HcneK are households in which the eldest child has not yet enrolled in kindergarten; HceK are households in which the eldest child has enrolled in kindergarten; HceEa are households in which the eldest child has enrolled in elementary school or above; Hcg are households with children and grandparents.

## 4. Discussion

The results of this study indicate that HceEas migrated from urban centers to suburban areas due to the spread of COVID-19 infection in the Japanese metropolitan areas. HceEas migrated not only because of the spread of the COVID-19 infection, but also due to the combined factors of children's schooling and parental caregiving.

HceEas were significantly more likely to place more importance on community and environments than on working arrangements, as Gustavus et al. [25] noted. In fact, before the pandemic all households placed more importance on work, including access to workplace. In addition, even during the pandemic, SHs were significantly more likely to migrate because of the change in employment/career. However, the results of this study imply that HceEas placed less priority on working because of changes in workstyle during the pandemic. HceEas were significantly more likely to work from home and utilize company support programs, such as the flextime working program. Therefore, this study's result is significant because it clarifies that HceEas did not prioritize working during the pandemic. In the United States, the Vermont government offers the New Remote Worker Grant for people working remotely or in co-working spaces [46]. Similar relocation programs have spread to many countries, such as Ireland [47]. Therefore, the changes in residential preference might be the same worldwide.

On the other hand, HceEas placed significantly higher importance on the ties to communities, the presence of the quality of childcare environment, and the presence of a large garden/balcony. The results suggest that during the urban exodus in Japanese metropolitan areas ties to communities, quality of childcare environment, and the presence of a large garden/balcony were the main elements of the "urban attractiveness" that Nathen et al. [9] pointed out. In regard to ties to communities, HceEas gave higher PD scores to migration during the pandemic. Therefore, HceEas prioritized ties to communities related to their children's schooling and their parental caregiving. Furthermore, HceEas migrated to places where they could feel secure in raising children and caring for their parents, rather than to familiar places with high PI scores from the beginning, such as places with access to their parents' houses. Kim et al. [32] noted that households with children chose their residential place based on a trade-off between access to workplaces and the richness of the natural environment. In regard to this trade-off during the pandemic in Japanese metropolitan areas, it was found that HceEas characterized their residential place based not only on the richness of the natural environment, but also on the tie to communities, the quality of childcare environment, and the presence of a large garden/balcony.

Moreover, HceEas were significantly more likely to utilize opportunities to connect with the community, such as trial migration support programs. This result suggests that municipalities wishing to encourage migration must offer trial migration support programs for HceEas, who value local communities. Furthermore, because these households have a strong sense of PD, establishing a support program for children's schooling and parental caregiving is likely to promote migration. In Italy, Portugal, Ireland, and Australia, rural relocation incentives are offered for migrants. Compared to the counties, the effectiveness of trial migration support programs might be unique in Japan [46]. In regard to migration, the Japanese might emphasize community more than other countries.

Additionally, HceEas placed particular importance on the quality of childcare environment; the result was the same for HcneKs and HceKs. In regard to the environment, it was found that many people placed importance on the natural environment in Poland [48]. In OECD countries, it was also reported that high-speed internet access and the natural environment were important [49]. Compared to these countries, Okada et al. [50] found that HceEas had emphasized the quality of the childcare environment before the pandemic in Japan. The importance of the quality of the childcare environment has not changed since before the pandemic. This result suggests that childcare and welfare support programs might effectively increase migration, even during the COVID-19 pandemic. The result might be unique in Japan, which is an aging society.

Finally, in terms of housing, HceEas were significantly more likely to buy and live in a new detached house for sale during the pandemic at the 1% level. In addition to HceEas, HceKs and HcneKs were significantly more likely to buy and live in a new detached house. HceEas, HceKs, and HcneKs were not significantly more likely to live in new detached houses for sale before the pandemic. For example, before the pandemic, HceEas suggested a significantly higher preference, at the 1% level, to live in public housing. The result was different from the result in South Korea [34]. Before the pandemic in Japan, Ono et al. [31] clarified that housing size and layout were the top residential priorities for households raising children. Therefore, households with children will continue to seek spacious housing, even after the COVID-19 pandemic.

## 5. Conclusions

In conclusion, this study elucidates that HceEas are those who migrate by way of the urban exodus during the COVID-19 pandemic in Japanese metropolitan areas. Regarding residential preferences, HceEas change the importance of community and environment rather than the working arrangement. As an academic contribution, this study clarified that HceEas tend to place more importance on the quality of childcare environment, ties to communities, the presence of a large garden/balcony, and utilizing opportunities to experience the community, such as via trial migration support programs. This result is novel and essential because it elucidates the role of household type and residence preference in the urban exodus in Japanese metropolitan areas. These results are summarized in Table 11. Table 11 listed significantly higher values at the 1% level by residual analysis in Tables 3–10.

In previous studies, Kato et al. [7] found that an urban exodus occurred in the Osaka metropolitan area during the COVID-19 pandemic. They elucidated that the population increased in the suburban area of Akashi City, Wakayama City, and Fukuchiyama City [7]. For example, Akashi City and Fukuchiyama City implemented childcare support programs [51,52]. In addition, Wakayama City actively implemented trial migration support programs [53]. Okada et al. [50] found that support programs encouraging ties to communities effectively promote migration in the Osaka metropolitan area. Therefore, as a policy implication, the results of this study verify that trial migration and childcare support programs might be triggers for HceEas, who prioritize ties to communities and the quality of childcare environment. Those programs might be effective in some countries facing the aging society, such as China and South Korea. Furthermore, they clarify that HceEas did not prioritize working arrangements, such as access to workplace. This flexible workstyle encouraged by the pandemic made it possible for HceEas to prioritize factors other than working, such as community and environment—the same as in other counties. This result is a novel conclusion that differs from the findings of previous studies conducted before the pandemic.

Based on the conclusions of this study, an additional research question arises: will the urban exodus continue after the COVID-19 pandemic? Two factors have caused the urban exodus: push factors, which decrease the population in urban centers, and pull factors, which increase the population in suburban areas. This study has clarified the pull factor that increases the population in suburban areas. During the early stages of the pandemic, push factors became social issues because urban centers were faced with serious problems, such as lockdowns. However, in Japan, most people have been vaccinated for SARS-CoV-2. In addition, we have developed a way to coexist with COVID-19 for the sake of socioeconomic sustainability. This means the push factor has begun to wane. Therefore, the urban exodus might stop, and people might begin to migrate to urban centers again. This study has elucidated that SHs continued to prioritize access to workplaces even during the COVID-19 pandemic. However, the results of this study indicate that the urban exodus is unlikely to stop, even after the pandemic. This is because workstyle and lifestyle have changed significantly in the wake of the COVID-19 pandemic. For example, many people worked from home during the pandemic. In addition, more and more companies have accepted various workstyles in order to attract the best workers [54].

In the social context, some households would probably choose their residential places based on their ties to communities and the quality of the childcare environment, rather than access to workplaces. In addition, more and more people in Japan are concerned about parental caregiving because of the aging society. The aging of society may cause the continued acceleration of the urban exodus in Japanese metropolitan areas. Therefore, HceEas might continue to migrate to attractive suburban areas. This result is significant because it suggests that the urban exodus might become a long-term urban phenomenon. Therefore, some municipalities in suburban areas might experience further population growth, even after the pandemic. The results contribute to developing effective migration policies for urban planners and administrative staff in not only Japan but also countries worldwide facing aging societies, such as China and some European countries.

A limitation of this study are the limited numbers of samples associated with the study period. Specifically, there are two limitations of this study. First, the samples are web-based questionnaire respondents who migrated during the COVID-19 pandemic (April 2020–March 2022). It is expected that some households wanted to migrate, but could not do so during the pandemic. Among the households that were able to migrate, the earliest adopters were likely the more socially advantaged. Therefore, the possibility of sample bias cannot be completely ruled out. In addition, we cannot preclude that the same results might be obtained after the pandemic. The second limitation is the analysis according to the pre-migration and post-migration addresses in the Osaka and Tokyo metropolitan areas. Due to the sample size of the web-based questionnaire, this study could only identify overall trends. However, detailed residential preferences might be different depending on the city types. The preferences might differ not only between metropolitan areas but also between cities within the same area. Further research needs to investigate the urban exodus with many samples—at minimum, a thousand samples.

**Table 11.** Summary of results.

| | | SH | MCH | HcneK | HceK | HceEa | Hcg |
|---|---|---|---|---|---|---|---|
| Triggers for migration. (in Table 3) | | • Change in employment/career | • Marriage • Retirement | • Marriage • Childbirth | • Childbirth | • Spread of the COVID-19 infection • Children's schooling • Parental caregiving | • Returning from abroad |
| Prioritization of residential preferences. (in Table 4) | | (During pandemic) Working | | | | (During pandemic) Community | |
| Detailed residential preferences. (in Table 5) | | | • Favorability of communities | • Neighborhood of relatives • Quality of childcare environment • Access to parents' homes | • Quality of childcare environment | • Tie to communities • Quality of childcare environment • Large garden/balcony | • Neighborhood of relatives |
| Place attachment. (in Table 6) | Place Identity Place Dependence | Down Down | | Up Up | Up | | |
| Housing type. (in Table 7) | | | (During pandemic) • Rent apartment • Employee's house | | (During pandemic) • New detached house for sale | (Before pandemic) • Public housing (During pandemic) • New detached house for sale | |
| Ratio of OF to WFH. (in Table 8) | | | | (During pandemic) OW:WFH = 0:100% | | (During pandemic) OW:WFH = 80:20% | |
| Company support systems. (in Table 9) | | | • Not applicable | • Childcare leave support programs | • Childcare leave support programs | • Flextime working programs | |
| Government support programs. (in Table 10) | | • Others | • Others | • Childcare support programs | • Childcare support programs | • Trial migration experience programs • Childcare support programs • Welfare support programs | |

**Author Contributions:** Conceptualization, M.K. and H.K.; methodology, M.K.; software, M.K.; validation, H.K.; formal analysis, M.K.; investigation, M.K.; resources, H.K.; data curation, M.K.; writing—original draft preparation, M.K.; writing—review and editing, M.K., H.K. and D.M.; visualization, M.K.; supervision, H.K.; project administration, M.K.; funding acquisition, H.K. All authors have read and agreed to the published version of the manuscript.

**Funding:** This research was funded by the JSPS KAKENHI (grant number 21K14318).

**Institutional Review Board Statement:** This research protocol was approved by the ethics committee of the Graduate School of Life Science, Osaka Metropolitan University (22-25).

**Informed Consent Statement:** Informed consent was obtained from all subjects involved in the study.

**Data Availability Statement:** The data presented in this study are available from the corresponding authors upon request.

**Conflicts of Interest:** The author declares no conflict of interest.

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
