# Peer review of "Why Did Urban Exodus Occur during the COVID-19 Pandemic from the Perspective of Residential Preference of Each Type of Household? Case of Japanese Metropolitan Areas"

_sustainability, doi:10.3390/su15043315_

Round 1
Reviewer 1 Report
In order to improve the research, I will make a series of recommendations to the authors:
In the summary, it is necessary for the authors to emphasize what the objectives of the study were and to briefly specify the analysis methods used.
In the introduction, the background chapter, the authors very vaguely explain the context of the study, referring to the behavior of residents from other states during the Covid 19 pandemic. The context related to Japan is not sufficiently explained. In order to understand the new preferences of Japanese residents during the pandemic, the reader must understand what their initial behavior was, who are the residents predisposed to migration (from the point of view of economic level, social belonging, age group and gender structure, residential area etc.).
In the introduction, the authors should present in detail, each residential area that is the object of the study, from the perspective of the characteristics related to the number of residents, the socio-economic structure of the population living in these metropolitan areas, characteristics related to the residential space, the economic level of the residents etc.
From a methodological point of view, the authors must justify the choice of sample areas. Are 593 questionnaires representative for the study considering the millionaire population of the sample areas, respectively the Osaka and Tokyo metropolitan areas? It is mandatory for the authors to describe the type of questionnaire, if it is structured or random, if they targeted from the beginning a certain percentage of the male/female population, certain age categories, social categories???
The literature review chapter must be expanded, referring to other studies that address residential migration during the Covid pandemic, place attachment in other countries.
Regarding the results of the study, the tests show the preferences of the residents in relation to different reasons, the running of the database not being done according to the city of origin.
Maybe it would have been necessary to compare the responses of Osaka residents with those of Tokyo or an approach in relation to age groups, etc. Also, the conclusions and discussions should address the issues stated above.
Author Response
Dear Reviewer:
We appreciate the reviewer for the generous comment on the manuscript. We have attached our response letter in PDF format. We believe that the manuscript is now suitable for publication in Sustainability and look forward to hearing from you concerning your decision.
Yours sincerely

Reviewer 2 Report
First, I would like to congratulate the authors for the work put into this article. Second, the following comments are formulated solely to improve the quality of the present manuscript, as follows:
1. The authors should consider entirely re-writing the abstract. Several words repetitions are present in the abstract ( e.g., this study, HceEas) making it hard to follow.
2. I suggest rephrasing the research question using a formal, academic form (lines 41-42).
3. "Besides this" should be modified in "besides" (line 38).
4. The crosstabulation tables should also report the percentages (or solely the percentages), same stands for the in-text references to the results.
5. Lines 399-405 from the conclusions section repeats the information presented in the discussion section and should be re-phrased.
Author Response

(The authors gave the same response as above.)

Reviewer 3 Report
The paper aims to investigate the reasons of urban exodus during the COVID-19 pandemic from the perspective of residential preference. The topic is very interesting, and the language of the paper is generally good. However, some major concerns need to be addressed before considering publication.
First, what's the research gaps in existing literature? and what's the major contributions of this work? The authors failed to clearly state the above points.
Second, the logic of the paper and connections between different parts are poor. For example, seven perspectives are analyzed in Section 3 Results. However, it is not clear why those aspects are analyzed in this study, and what's the links between different parts of the results?
Third, the discussion parts need to be largely improved by including more international perspectives. What's the policy implications of your work to other regions or countries? What's the difference of your work compared to studies in other regions or countries?
Fourth, the authors may consider the usage of figures to present their results and main methods or findings. The quality of tables also needs to be improved.
Author Response

(The authors gave the same response as above.)

Round 2
Reviewer 1 Report
The work has been considerably improved, the current approach being much more coherent and clear.
Congratulations to the authors!
Reviewer 2 Report
Article properly revised.